# The impact of age on genetic risk for common diseases

Xilin Jiang[1,2,3], Chris Holmes[1,2,4], Gil McVean[1]*

**1** Big Data Institute, Li Ka Shing Centre for Health Information and Discovery, University of Oxford, Oxford, United Kingdom, **2** Department of Statistics, University of Oxford, Oxford, United Kingdom, **3** Wellcome Centre for Human Genetics, University of Oxford, Oxford, United Kingdom, **4** The Alan Turing Institute, London, United Kingdom

* gil.mcvean@bdi.ox.ac.uk

**Data Availability Statement:** This research has been conducted using the UK Biobank Resource; application number 12788. This work uses data provided by patients and collected by the NHS as part of their care and Support. The code generated

## Abstract

Inherited genetic variation contributes to individual risk for many complex diseases and is increasingly being used for predictive patient stratification. Previous work has shown that genetic factors are not equally relevant to human traits across age and other contexts, though the reasons for such variation are not clear. Here, we introduce methods to infer the form of the longitudinal relationship between genetic relative risk for disease and age and to test whether all genetic risk factors behave similarly. We use a proportional hazards model within an interval-based censoring methodology to estimate age-varying individual variant contributions to genetic relative risk for 24 common diseases within the British ancestry subset of UK Biobank, applying a Bayesian clustering approach to group variants by their relative risk profile over age and permutation tests for age dependency and multiplicity of profiles. We find evidence for age-varying relative risk profiles in nine diseases, including hypertension, skin cancer, atherosclerotic heart disease, hypothyroidism and calculus of gallbladder, several of which show evidence, albeit weak, for multiple distinct profiles of genetic relative risk. The predominant pattern shows genetic risk factors having the greatest relative impact on risk of early disease, with a monotonic decrease over time, at least for the majority of variants, although the magnitude and form of the decrease varies among diseases. As a consequence, for diseases where genetic relative risk decreases over age, genetic risk factors have stronger explanatory power among younger populations, compared to older ones. We show that these patterns cannot be explained by a simple model involving the presence of unobserved covariates such as environmental factors. We discuss possible models that can explain our observations and the implications for genetic risk prediction.

## Author summary

The genes we inherit from our parents influence our risk for almost all diseases, from cancer to severe infections. With the explosion of genomic technologies, we are now able to use an individual's genome to make useful predictions about future disease risk. However,

during this study is available at https://github.com/Xilin-Jiang/longitudinal_genetic_analysis.

**Funding:** Funded by Wellcome (BST00080-H503.01 to XJ, 100956/Z/13/Z to GM, https://wellcome.org); the Li Ka Shing Foundation (to GM, https://www.lksf.org); The Alan Turing Institute (https://www.turing.ac.uk), Health Data Research UK (https://www.hdruk.ac.uk), the Medical Research Council UK (https://mrc.ukri.org), the Engineering and Physical Sciences Research Council (EPSRChttps://epsrc.ukri.org) through the Bayes4Health programme Grant EP/R018561/1, and AI for Science and Government UK Research and Innovation (UKRI, https://www.turing.ac.uk/research/asg) (to CH).The funders had no role in study design, data collection and analysis, decision to publish, or preparation of the manuscript.

**Competing interests:** G.M. is a director of and shareholder in Genomics PLC, and is a partner in Peptide Groove LLP. The other authors declare no competing financial interests.

recent work has shown that the predictive value of genetic information varies by context, including age, sex and ethnicity. In this paper we introduce, validate and apply new statistical methods for investigating the relationship between age and the contributions of genetic risk. These methods allow us to ask questions such as whether relative risk is constant over time, precisely how relative risk changes over time and whether all genetic risk factors have similar age profiles. By applying the methods to data from the UK Biobank, a prospective study of 500,000 people, we show that there is a tendency for genetic relative risk to decline with increasing age. We consider a series of possible explanations for the observation and conclude that there must be processes acting that we are currently unaware of, such as distinct phases of life in which genetic risk manifests itself, or interactions between genes and the environment.

## Introduction

Many studies have demonstrated the potential utility of using genetic risk factors for predicting individual risk of common diseases, ranging from heart disease [1,2] to breast cancer [3] and auto-immune conditions [4]. Genetic risk coefficients can be estimated from cross-sectional genome-wide association studies, which estimate enrichment of common genetic variants among clinically-ascertained (or sometimes self-reported) cases. Genome-wide scores, typically referred to as polygenic risk scores (PRS), are usually constructed as linear combinations of individual variant effects, though there is considerable variation in how variants are selected for inclusion and how coefficients are estimated [5]. Nevertheless, validation on independent data sets has demonstrated odds-ratios for PRSs that are comparable to established risk factors, both lifestyle-related [6] and monogenic [7], thus providing an impetus for their adoption within health management, both at individual and population levels; though one study has suggested that PRS does not provide additional prediction power over clinical risk factors for coronary heart disease [6].

One aspect of genetic risk estimation that has not been fully explored is the role of age in modulating the effects of genetic risk. Several studies have shown that the prediction power of PRS varies across age groups in diseases including breast cancer [8], ischaemic heart disease [9] and prostate cancer [10]. Similarly, the standardised incidence ratio of breast cancer for *BRCA1/2* mutation carriers also varies with age [11]. Moreover, genetic analyses of quantitative traits including blood pressure, lipid levels and BMI have identified genetic variants whose effect size changes with age [12–16]. These results raise the possibility that genetic risk factors may play larger or smaller roles in influencing risk of disease during different age intervals. However, the longitudinal analysis of disease risk has to account appropriately for the impact of selection that arises in age-stratified analyses; even under a time-invariant proportional hazards model, those entering the disease state earliest will tend to be those with the highest burden of risk factors. This can be particularly problematic when not all risk factors are measured, as hidden risk factors can act to apparently dilute relative risk over time [17–19].

Here, we address two open questions in the analysis of longitudinal genetic risk for common disease. First, we introduce a method to infer the nature of the relationship between age and genetic relative risk for individual variants that is appropriate for censored data such as that available from biobanks. Because the information available for single variants is relatively weak, we use a Bayesian clustering approach to identify sets of variants that show similar profiles of risk with age. On applying the method to data from the UK Biobank on 24 common diseases, our primary finding is that, in agreement with previous observations, for age-varying

genetic risk profiles, genetic factors most strongly influence risk of early disease. However the quantitative nature of the relationship between genetic relative risk and age varies among diseases and, for some, we find evidence for multiple, distinct profiles of age-varying risk. Second, we consider whether observed patterns can potentially be explained by the presence of unmeasured risk factors. To achieve this, we fit parametric models that accommodate unmeasured variation in risk to age-varying incidence and use these models to predict the drop-off in apparent genetic relative risk that could be attributed to such a phenomenon, known as frailty. We find that the observed drop-off in risk has a qualitatively different profile from that expected from simple models of frailty. Rather, our observations indicate that genetic relative risks (conditional on all other risk factors) vary with age.

## Results

### Data description

We used the genotype data, individual information and Hospital Episode Statistics (HES) data from the UK Biobank dataset [20]. We identified 24 disease-specific ICD-10 codes (Tables 1 and S1) with a prevalence > 0.5% in the entire cohort and for which at least 20 independent associated variants were identified using the TreeWAS model [21]. To reduce the impact of confounding, we focused analyses on the 409,694 individuals of British Isles ancestry (188,268 men and 221,426 women with an average age at recruitment of 66.5 years), though additional analyses were carried out on the entire cohort and other ethnic groups. We performed additional analyses to evaluate the robustness of our observation when generalised to other self-identified ethnicities, sets of associated traits, and disease definitions. To compare our analysis within different ethnic groups, we analysed individuals from all ethnic backgrounds (N = 501,756), those identifying as Black or Black British (N = 8,039) and those identifying as South Asian (N = 8,024). To compare our analysis using different disease ontology systems, we used phecodes [22] to map the selected ICD10 codes to phenotypes and inferred genetic risk profiles using collapsed traits and variant sets. To compare our results using different set of variants, we used variants reported in the GWAS Catalog [23] for two traits, "hypertension" and "coronary heart disease", and inferred genetic risk profiles on the respective ICD-10 codes, I10 and I25.1.

### Age-profiles for genetic risk scores

To first demonstrate that age-varying genetic risk is a common feature of complex disease we estimated genetic risk coefficients through logistic modelling of a training case-control study (across all ages) and then assessed the efficacy of a combined genetic risk score to differentiate between cases and controls within each age group in an independent testing set (see S1 Supplemental Methods and Fig 1). For many diseases, and notably those identified later as having statistically significant evidence for non-uniform genetic relative risk profiles, we found a typically decreasing risk profile. (Figs 2 and S1) For example, the odds-ratio for the 90th percentile of GRS for I25.1 "atherosclerotic heart disease of native coronary artery" drops from 3.63 [3.47, 3.792] in the youngest age group to 1.77 [1.668, 1.88] in the oldest. We also note while some disorders, such as E78.0 "pure hypercholesterolemia", show a very dramatic decrease in risk between the two youngest age groups (Odds Ratio of top GRS decile: 2.209 [2.121, 2.297] to 1.633 [1.576, 1.69]), others, such as a I10 "essential (primary) hypertension", show a much more gradual decline (Odds Ratio of top GRS decile: 1.513 [1.474, 1.552] to 1.512 [1.483, 1.541]). These results suggest that the relationship between genetic risk and age varies among diseases and may indeed vary among variants, and motivates a more principled approach to the analysis of such data. Estimates of odds ratios for the 90th and 80th percentile of the GRS distribution are provided in S2 Table.

**Table 1. Summary of evidence for age-varying genetic risk by disease.**

| ICD-10 code | Description | Prevalence in UK Biobank | Number of associated variants | Mean reported age of onset | P1 | Q1 | P2 | Q2 | P3 | Q3 |
|---|---|---|---|---|---|---|---|---|---|---|
| C44.3 | Other and unspecified malignant neoplasm of skin of other and unspecified parts of face | 1.41% | 34 | 62.64 | 0.0159* | 0.0633* | 0.0739 | 0.2531 | 0.0092* | 0.1093 |
| C44.5 | Other and unspecified malignant neoplasm of skin of trunk | 0.48% | 25 | 61.45 | 0.0016** | 0.0091** | 0.0143* | 0.0853* | 0.5644 | 0.7964 |
| C78.7 | Secondary malignant neoplasm of liver and intrahepatic bile duct | 0.64% | 30 | 64.20 | 0.3753 | 0.5124 | 0.6920 | 0.8368 | 0.5973 | 0.7964 |
| C79.5 | Secondary malignant neoplasm of bone and bone marrow | 0.52% | 29 | 64.56 | 0.4649 | 0.5579 | 0.8287 | 0.8647 | 0.3122 | 0.6242 |
| E03.9 | Hypothyroidism, unspecified | 3.46% | 32 | 60.61 | 0.0329* | 0.0876* | 0.0592 | 0.2501 | 0.8010 | 0.8738 |
| E10.9 | Type 1 diabetes mellitus without complications | 0.63% | 28 | 57.32 | 0.4975 | 0.5686 | 0.7828 | 0.8539 | 0.0566 | 0.2353 |
| E11.9 | Type 2 diabetes mellitus without complications | 4.38% | 75 | 61.95 | 0.4103 | 0.5182 | 0.5358 | 0.7610 | 0.2710 | 0.6242 |
| E66.9 | Obesity, unspecified | 2.46% | 21 | 60.71 | 0.0981 | 0.2067 | 0.2609 | 0.5350 | 0.5420 | 0.7964 |
| E78.0 | Pure hypercholesterolemia | 8.12% | 43 | 62.59 | 0.0001** | 0.0001** | 0.0001** | 0.0001** | 0.0449* | 0.2353 |
| I10 | Essential (primary) hypertension | 18.98% | 80 | 61.87 | 0.0001** | 0.0001** | 0.0001** | 0.0001** | 0.3764 | 0.6604 |
| I20.0 | Unstable angina | 1.26% | 46 | 59.92 | 0.9021 | 0.9021 | 0.7323 | 0.8368 | 0.0356* | 0.2353 |
| I20.9 | Angina pectoris, unspecified | 3.84% | 78 | 61.84 | 0.0244* | 0.073* | 0.1269 | 0.3382 | 0.1902 | 0.6242 |
| I21.9 | Acute myocardial infarction, unspecified | 0.92% | 24 | 62.08 | 0.7556 | 0.7884 | 0.3121 | 0.5350 | 0.6356 | 0.8028 |
| I25.1 | Atherosclerotic heart disease of native coronary artery | 4.46% | 116 | 61.57 | 0.0001** | 0.0001** | 0.0001** | 0.0001** | 0.0001** | 0.0001** |
| I25.2 | Old myocardial infarction | 1.72% | 83 | 64.32 | 0.0096* | 0.0456* | 0.1053 | 0.3156 | 0.2198 | 0.6242 |
| I25.9 | Chronic ischaemic heart disease, unspecified | 2.66% | 86 | 63.69 | 0.1285 | 0.2202 | 0.4428 | 0.7084 | 0.2452 | 0.6242 |
| I50.1 | Left ventricular failure, unspecified | 0.83% | 22 | 63.22 | 0.0696 | 0.1669 | 0.2971 | 0.5350 | 0.9950 | 0.9950 |
| J44.9 | Chronic obstructive pulmonary disease, unspecified | 1.78% | 24 | 64.40 | 0.3329 | 0.5124 | 0.5391 | 0.7610 | 0.8452 | 0.8819 |
| J45.9 | Other and unspecified asthma | 6.34% | 35 | 57.54 | 0.1056 | 0.2067 | 0.2027 | 0.4863 | 0.5087 | 0.7964 |
| K29 | Gastritis and duodenitis | 7.02% | 33 | 58.87 | 0.3843 | 0.5124 | 0.7056 | 0.8368 | 0.0589 | 0.2353 |
| K80.2 | Calculus of gallbladder without cholecystitis | 2.11% | 26 | 57.99 | 0.0236* | 0.073* | 0.0626 | 0.2501 | 0.7942 | 0.8738 |
| M06.9 | Rheumatoid arthritis, unspecified | 0.99% | 54 | 60.16 | 0.3548 | 0.5124 | 0.6271 | 0.8361 | 0.3853 | 0.6604 |
| M19.9 | Osteoarthritis, unspecified site | 3.70% | 22 | 62.80 | 0.1120 | 0.2067 | 0.2981 | 0.5350 | 0.2989 | 0.6242 |
| M54.5 | Low back pain | 1.90% | 29 | 56.75 | 0.6320 | 0.6894 | 0.8935 | 0.8935 | 0.7793 | 0.8738 |

P1: permutation test for fitting a linear profile over age; P2: permutation test for fitting a quadratic polynomial profile over age; P3: permutation test for multiple profiles over age. ** P < 0.005, * P < 0.05
Q1: FDR adjusted P1 value; Q2: FDR adjusted P2 value; Q3: FDR adjusted P3 value. ** Q < 0.01, * Q < 0.1

Summary of ICD-10 disease codes analysed and evidence for age-varying effect sizes and number of age-profile classes. "Prevalence in UK Biobank" is the proportion of the British Isles ancestry subgroup that has at least one record of the ICD-10 code.

## Statistical inference of age-varying genetic risk with multiple variant categories

To estimate age-specific effects of variants we divided age into eight intervals and used an interval-censoring approach in which we applied a Cox Proportional Hazard model to subjects at risk within each age interval. Specifically, the hazard rate for the risk factor is estimated by comparing those whose first disease event occurs during the interval in question to those who

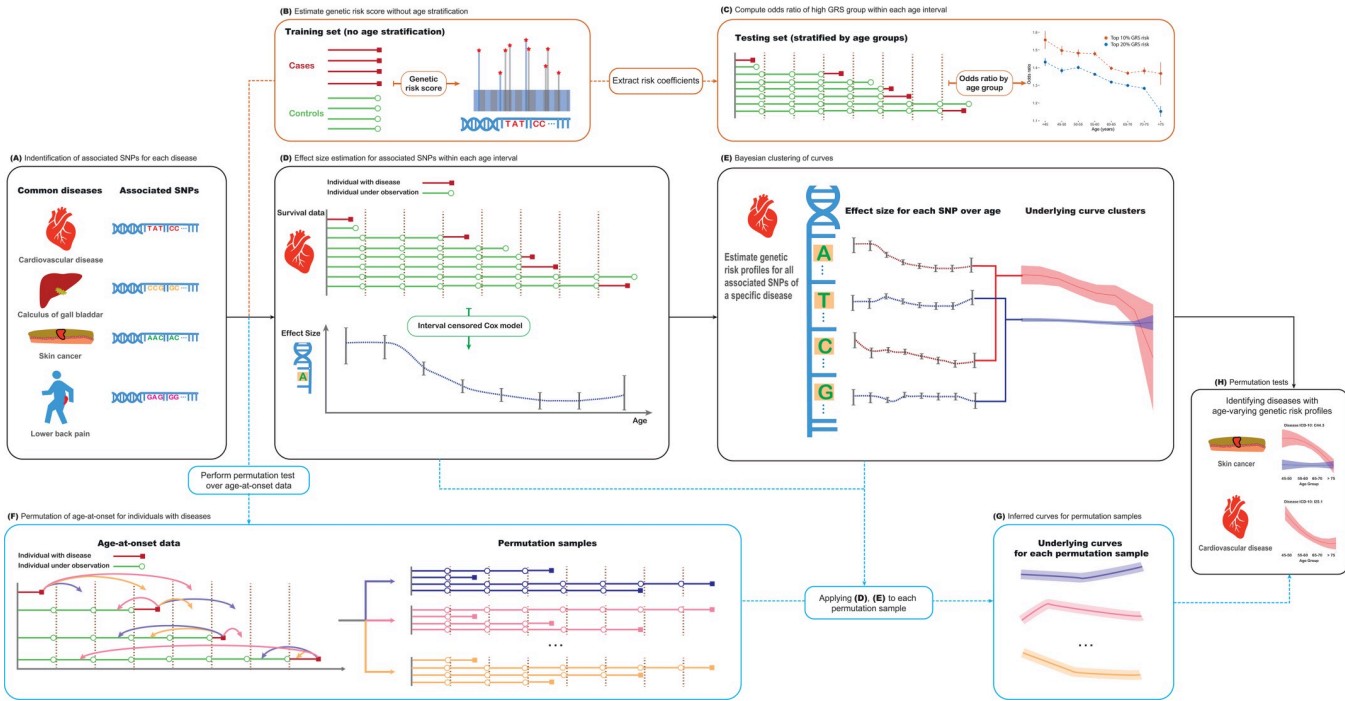

**Fig 1. Schematic representation of methodology.** (A) Independent variants associated with a trait of interest are identified by analysis of the entire UK Biobank cohort using the TreeWAS methodology [40]. (B) Logistic regression is applied to estimate coefficients for variants on each trait using the training set. (C) Coefficients are used to compute individual genetic risk scores; the odds ratio associated with high GRS within each age group are estimated in the testing set. (D) An interval-censored proportional hazards model [44] is used to estimate the effect (and associated standard error) of each variant on the trait of interest within each of eight age intervals. (E) Bayesian clustering is used to estimate population age-profiles of risk, using either linear models or quadratic polynomials to encourage smoothness. (F-H) Permutation is used to test for age-homogeneity of effect size as well as to assess the evidence for multiple age profiles.

have a non-disease censoring event during the interval (such as death from a different disease, or drop-out from the study for reasons unrelated to disease) along with those who have neither a disease nor a censoring event during the interval (Fig 1). For a given variant, we estimated the effect size and its standard error for each interval using a proportional-hazards approach, using case-control matching to control for additional covariates such as date of birth, sex, BMI and 40 genetic principal components (see S1 Supplemental Methods). Effect sizes for individual SNPs were estimated in both univariable and multivariable settings (see below). Because estimated variant-interval coefficients have high uncertainty, we used a Bayesian clustering approach to estimate latent profiles of age-specific genetic risk, encouraging smoothness of profiles through spline functions. Finally, to test for deviations from homogeneity of risk over age, and to test for the presence of multiple age-specific risk profiles, we used a permutation strategy. Multiple risk profiles for a particular disease can occur when subsets of genetic factors associate with distinct age-varying patterns, for example, some factors may exhibit no variation with age while others show decreasing risk with age. Full details of the methods are given in the S1 Supplemental Methods and S1 Analytical Note.

To evaluate the methodology under the assumptions of the fitted model, we used stochastic simulation, varying the number of distinct profiles and their departure from uniformity. We first considered a likelihood ratio test (LRT) approach, fitting a linear model for risk profiles over age. Under realistic assumptions about the magnitude of effect sizes and number of associated variants we found that the multivariable approach is well-calibrated in its rejection of the null model of uniformity (i.e. when effect sizes are constant over time the LRT test has a

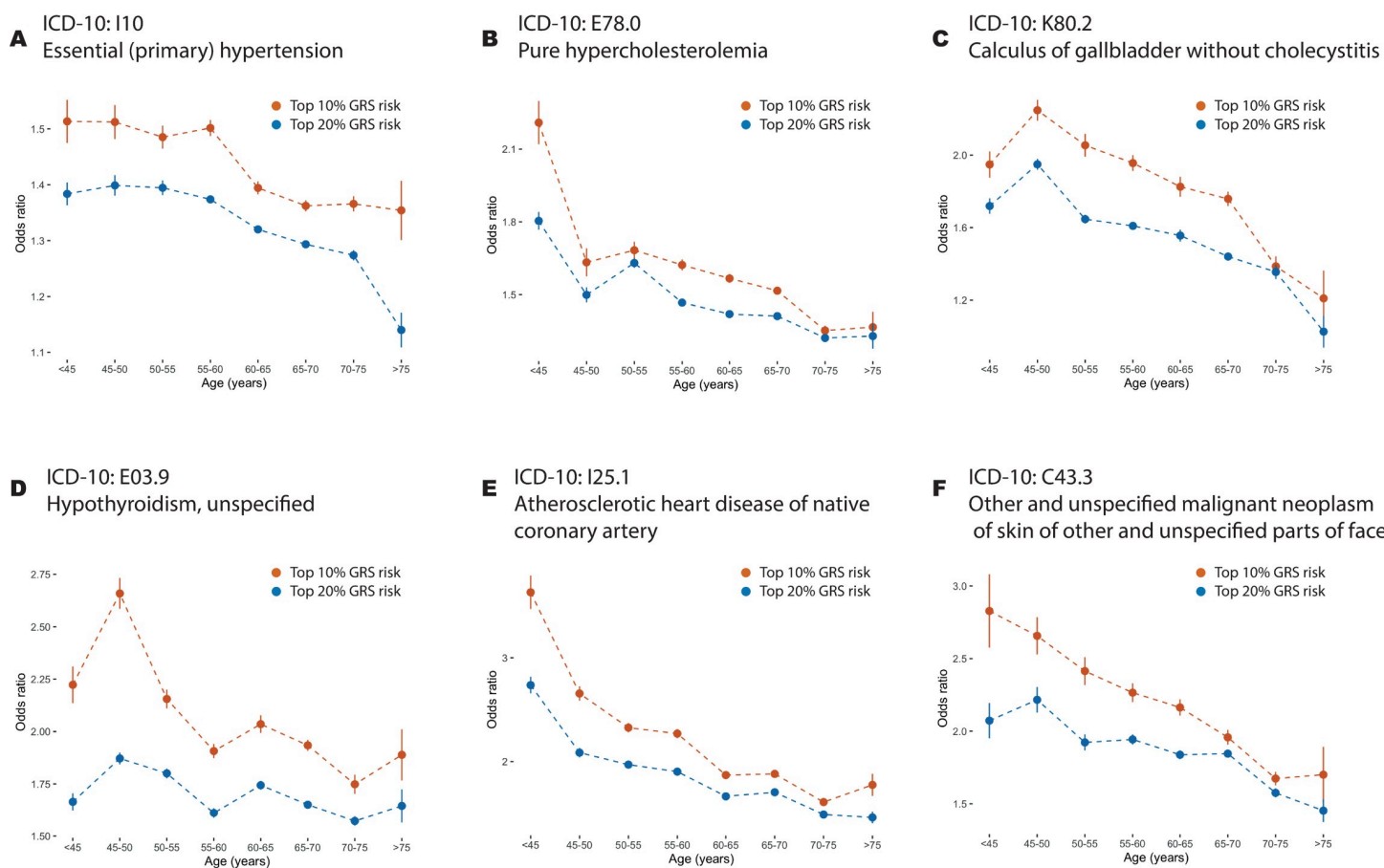

**Fig 2. Age-stratified odds-ratios for combined genetic risk scores.** (A-F) Age-stratified odds ratios in held-out testing data for genetic risk scores for six disorders where there is evidence for a single non-constant genetic risk profile, "Primary (essential) hypertension" (ICD-10 code I10), "pure hypercholesterolaemia" (E78.0); "Calculus of gallbladder without cholecystitis" (K80.2) and "Hypothyroidism, unspecified" (E03.9); "atherosclerotic heart disease of native coronary artery" (I25.1) and "other and unspecified malignant neoplasm of skin and unspecified parts of face" (C44.3). Results for all diseases are shown in S1 Fig. Odds ratios for the 80th (blue) and 90th percentiles of a combined genetic risk score within matched case-control samples (four controls for each case) are shown for each age interval; points indicate the average odds ratio of twenty five-fold cross-validation analyses with lines indicating the 95% confidence interval.

false positive rate of 0.048 at $P \leq 0.05$). When effect sizes are the same for all variants but these change by at least 0.6% per year (either increasing or decreasing), our approach has over 90% power to reject uniformity (Fig 3A). When quadratic polynomials were used to capture a wider range of possible risk profiles, we found that the LRT was less well calibrated under the null (false positive rate of 0.0725 at $P \leq 0.05$; Fig 3A), hence we adopted a permutation strategy for analysing empirical data. When applying the quadratic model to data simulated under a linear profile, we find a good match between true and inferred profiles (Fig 3B).

To simulate multiple cluster profiles, we modelled 10% of the variants as having a shared linear slope (the remainder being constant over age) and used a LRT to assess the evidence for multiple risk profiles. Here, we found that a 4% per year change in risk was required to achieve 90% power (at $P \leq 0.05$) to detect multiple clusters (Fig 3C). Under the null (all variants have a constant profile) the test has a false positive rate of 0.063 for the linear and 0.088 for the quadratic polynomial fitting at $P \leq 0.05$. When using the quadratic model to fit risk profiles we find a good match between true and inferred profiles (Fig 3D). We therefore conclude that the approach has sufficient power to detect deviations from constant profiles and provide unbiased

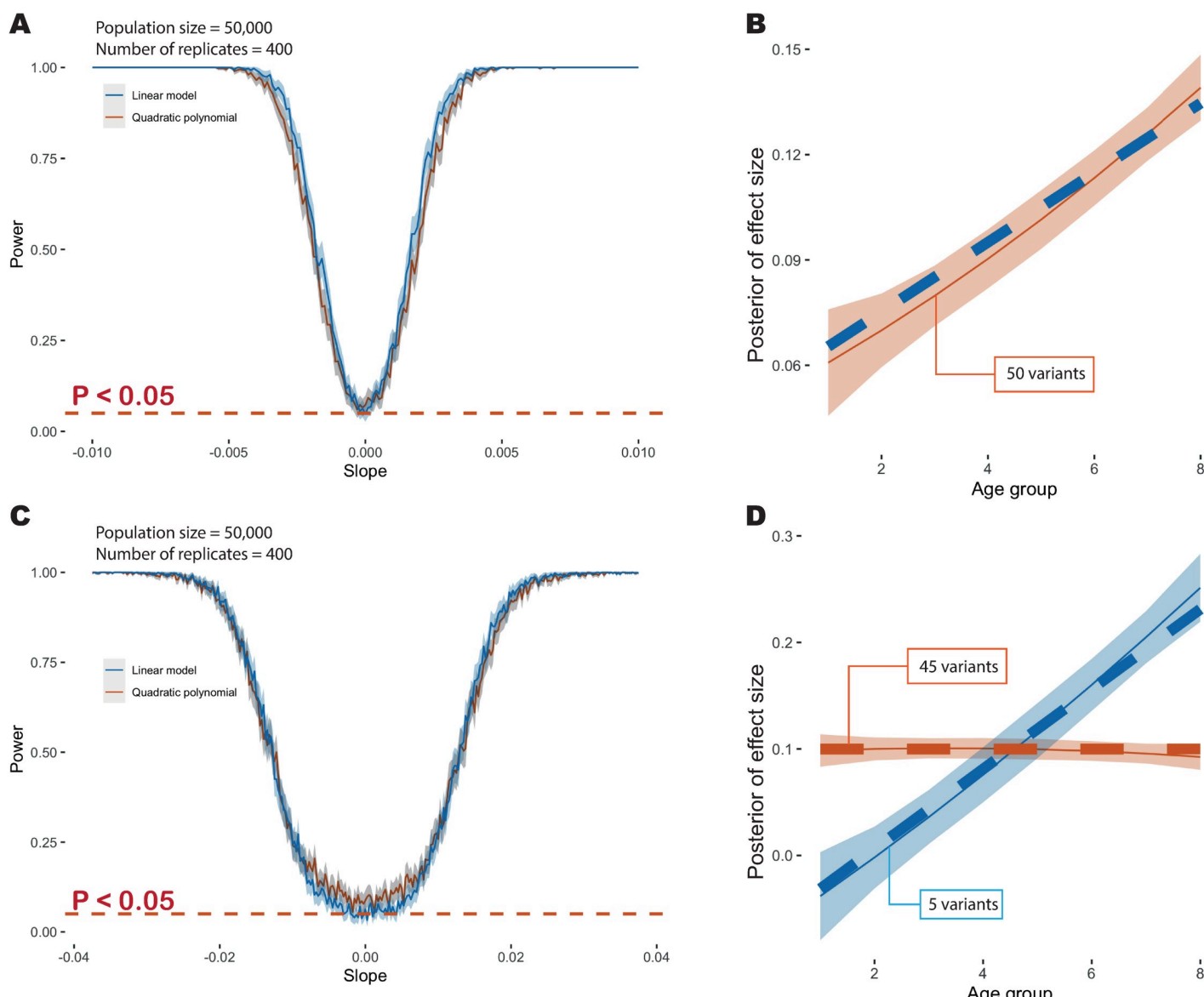

**Fig 3. Overview of simulation results.** (A) Power at P ≤ 0.05 to detect deviation from age-homogeneity as a function of slope in a model where effect sizes change linearly with age. The blue line indicates the point estimate when using a linear model to fit, the red line indicates the point estimate with a quadratic polynomial model and the grey shading indicates the 95% confidence interval. (B) Example showing the age-profile under which data are simulated (dashed blue line) and the inferred age profile (dashed red line) and 95% credible interval (red shading). (C) Power at P ≤ 0.05 to detect multiple age profiles in a simulation where 90% of variants have a time-invariant profile and 10% have an effect size that increases with age. The solid blue line indicates power when fitting a linear model and the solid red line indicates power when fitting a quadratic model. The dashed red line indicates the nominal significance threshold. Note the change in x-axis scale compared to Fig 2A. (D) Example showing inferred age-profiles for the two components (mean posterior and 95% credible interval). Additional simulation details are provided in the S1 Supplemental Methods and S2 Fig.

estimates of risk profiles in data sets of comparable size and complexity to the UK Biobank. When analysing multiple diseases we used a FDR approach to correct for multiple testing.

## Application to common diseases in the UK Biobank

To formally consider evidence for a non-linear relationship between genetic risk and age for the 24 diseases in Table 1, we applied the novel methods outlined above. When effects for

variants are estimated jointly and fitted to a linear latent profile, we identified, through permutation, nine diseases with evidence (P < 0.05) of a departure from constant genetic risk over age (Table 1). These are: C44.3 "other and unspecified malignant neoplasm of skin of other and unspecified parts of face"; C44.5 "unspecified malignant neoplasm of skin of trunk"; E03.9 "hypothyroidism, unspecified"; E78.0 "pure hypercholesterolemia"; I10 "essential (primary) hypertension"; I20.9 "angina pectoris, unspecified"; I25.1 "atherosclerotic heart disease of native coronary artery"; I25.2 "Old myocardial infarction" and; K80.2 "calculus of gallbladder without cholecystitis". All diseases have Q < 0.1 after FDR analysis. To model non-linearity we compared polynomial and cubic spline models with different degrees of freedom (S3 Fig) and selected the quadratic polynomial model using likelihood ratio tests. No additional diseases were identified as having non-constant risk profiles when fitting a quadratic polynomial and only four of the original nine (E78.0, I10, I25.1 and C44.5) remain significant (Table 1). However, we find one additional disease (I20.0 "unstable angina") and three of the above diseases (C44.3, E78.0 and I25.1) show evidence for more than one age-related risk profile (P < 0.05; Table 1, though only I25.1 has Q < 0.1).

As in the genetic risk score analysis, a common feature of the estimated risk profiles over age is a trend towards smaller effect sizes with increasing age (Fig 4A, 4B, 4C and 4D). For example, for I25.1, we find posterior of effect size drops by 50% from 45 years old to 75 years old and for C44.5 we find the posterior drops by 58% over the same interval. (S3 Table). Where diseases may have multiple risk profiles (Fig 4E and 4F), at least one of these is also typically decreasing with age. Profiles for all 24 diseases are shown in S4 and S5 Figs. We find no compelling examples of increasing risk over age. These results are consistent with the effects of genetic risk factors to have a larger impact on the risk of early disease [16,24], rather than late disease, though it is important to note that the absolute rate of disease typically increases with age for all diseases studied here. Estimates of genetic risk profiles (under a model of one variant class) are provided in S4 Table. We found that the decreasing pattern is largely consistent when estimating genetic risk profiles using the entire UK Biobank regardless of ethnic background, individuals who self-identify as Black or Black British, and those who self-identify as South Asians (S6 and S7 Figs). However, we found an increasing risk profile in one disease (P = 0.006), J45.9 "other and unspecified asthma", for individuals identifying as Black or Black British.

To examine robustness to the design of our analysis, we considered two extensions. First, when inferring the risk profiles for I10 "essential (primary) hypertension" and I25.1 "atherosclerotic heart disease of native coronary artery" we analysed variants reported in the GWAS Catalog for "hypertension" and "coronary artery disease" respectively. This showed similar decreasing patterns (but of different magnitude) compared to those inferred from the variants identified by TreeWas (S8 Fig). Second, we used phecodes to combine ICD-10 codes that, where appropriate, represent similar phenotypes. In line with the single-code results, we found decreasing risk profiles for 172.20" (other non-epithelial cancer of skin, ICD-10 codes: C44.3 and C44.5) and "411.20" (myocardial infarction, ICD-10 codes: I21.9 & I25.2).

One corollary of the decreasing genetic risk is that the GRS estimated within younger populations should have more power to discriminate between cases and controls within populations of similar ages. To demonstrate this, we evenly divided the cases into a younger group and an older group by age-at-onset and estimated GRS within each age group (See S1 Supplemental Methods). Using five-fold cross-validation, we computed and compared the average areas under curve (AUC) for both age groups using their age-stratified GRS (S9 Fig and S5 Table). We found the AUC differences between younger and older groups agree with the decreasing slopes for the diseases considered (S3 Table). For example, C44.5 "Other and unspecified malignant neoplasm of skin of trunk" has the steepest slope among the 24 codes and the AUC

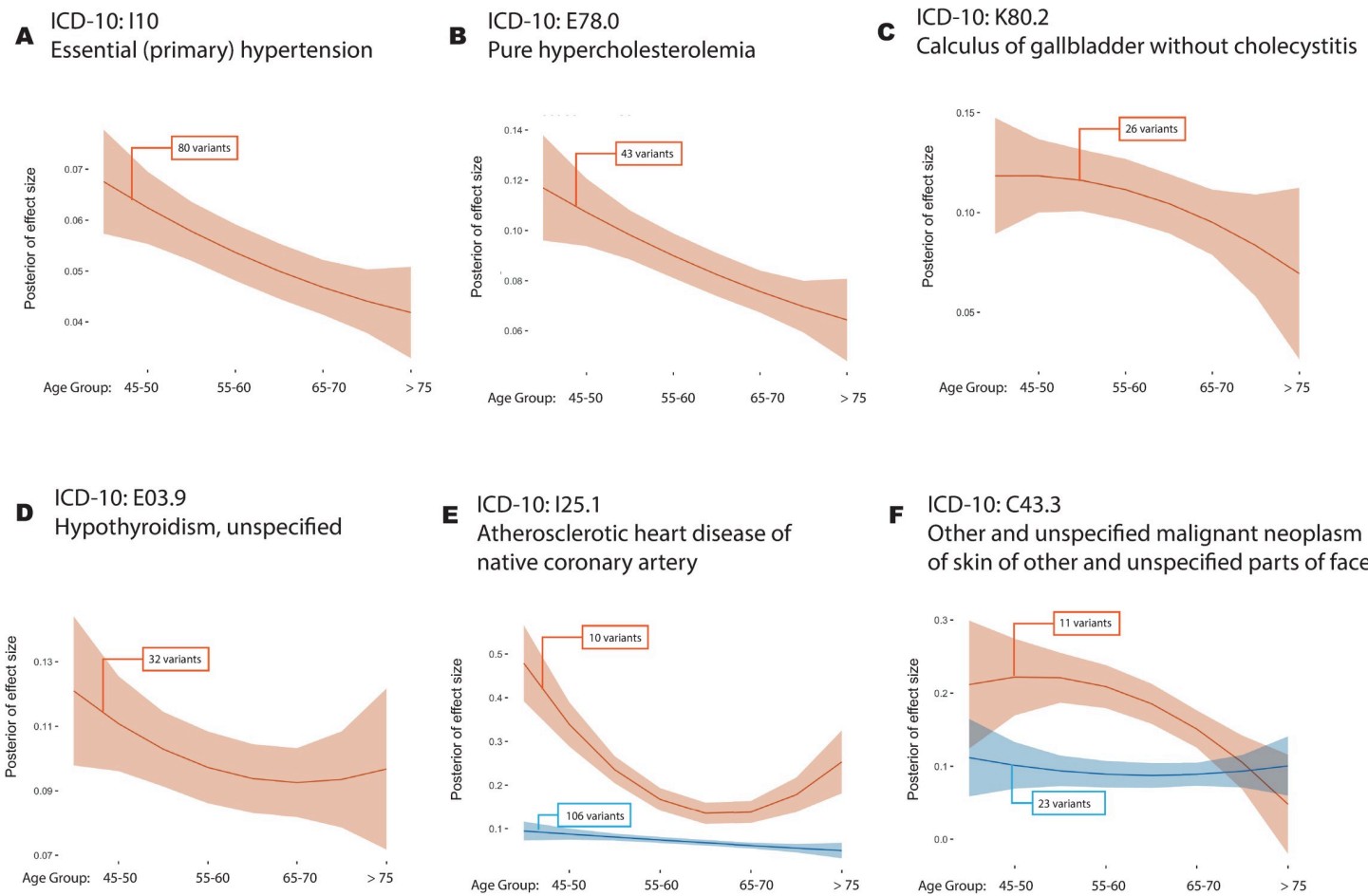

**Fig 4. Age-varying disease risk profiles.** (A-D) Inferred cluster profiles for four disorders where there is evidence for single non-constant profile; "Primary (essential) hypertension" (ICD-10 code I10; P = 0.0001), "pure hypercholesterolaemia" (E78.0; P = 0.0001), "Calculus of gallbladder without cholecystitis" (K80.2; P = 0.0236) and "Hypothyroidism, unspecified" (E03.9, P = 0.0329); (E-F) Inferred cluster profiles for two disorders where there is evidence for multiple non-constant profiles; "atherosclerotic heart disease of native coronary artery" (I25.1; P = 0.0001) and "other and unspecified malignant neoplasm of skin and unspecified parts of face" (C44.3; P = 0.0092). Curves for all diseases are shown in S4 Fig; Curves for all UK Biobank subjects regardless of ethnic background and for subjects from Black or South Asian ethnic background are shown in S6 Fig. The solid line indicates the posterior mean and the shaded area the 95% credible interval; Numbers in boxes indicate the number of variants in each cluster; All estimates are made with quadratic models for age-varying risk profiles.

difference (0.058) is also the largest. The only disease which has a larger AUC in the older group is M54.5 "low back pain", which also records the only increasing genetic risk over age, though the increasing pattern is not significant. Additionally, we applied the GRS estimated from the younger or older age group to compute the ROC in the other age group. In 18 out of 24 ICD-10 codes, the GRS estimated in the younger group has a bigger AUC than the GRS estimated in the older group, when applied to the testing set from the older population. We found no compelling differences in the discriminating power of GRS estimated from younger and older groups when applied to the testing set from the younger populations.

## The impact of unobserved risk factors

One possible explanation for the decreasing impact of genetic risk is the presence of unobserved risk factors. For any causal covariate of interest, the presence of unmeasured and causally-associated uncorrelated covariates has the effect of generating (at the population level) additional variability in hazard rates. Such heterogeneity, historically referred to as frailty in epidemiology

[19], has the potential to induce bias in effect sizes over time, somewhat remarkably even if independent of the covariate of interest, due to the increased rate at which individuals with high unmeasured risk enter into a disease state. Over time, those individuals with a risk-increasing covariate, but who do not have the disease, will become enriched for a protective background. Frailty will thus tend to lead to an underestimate of true effect sizes in older populations and, consequently, can even lead to biased effect size estimates (typically underestimates) in regression analysis of the entire cohort [18]. To demonstrate the impact of such covariates we repeated the simulations under a constant risk profile, but multiplied individual risk by an unobserved factor that is generated from a gamma distribution (Fig A in S2 Fig). We find that our test for age-dependence has a false positive rate of above 0.05 if the variance in risk is greater than 0.1 of the mean (Fig B in S2 Fig; specifically FPR > 0.09 when variance > 0.1 x mean).

To investigate the extent to which unmeasured genetic factors might be responsible for the diminishing of risk over time we first compared the results of univariable and multivariable analyses of the variants analysed here (Fig 5A). We found that results were essentially identical under the two approaches, suggesting the frailty arising from variants included here cannot explain the pattern. We next attempted to estimate general parameters of frailty using incidence data from the UK Biobank by fitting a parametric model in which the underlying disease incidence (baseline hazard rate) increases in proportion to age as a power function of age, but where there is a distribution of rates within the population, parameterised as a gamma distribution with a mean of one and an unknown variance [17,25]; see S1 supplemental Methods and S1 Analytical Note. Estimates of parameters are provided in S6 Table, along with the significance value for a goodness-of-fit test for the inferred model. We find substantial variation across diseases in the inferred

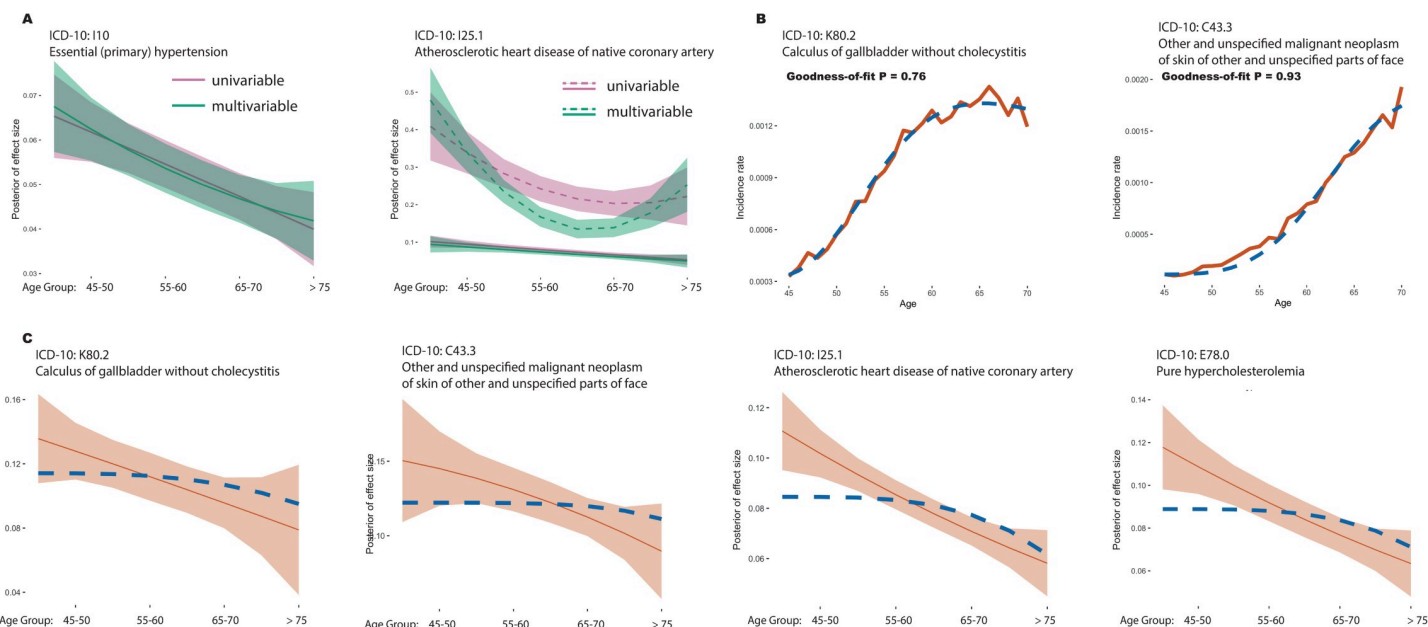

**Fig 5. The impact of frailty on genetic risk profiles.** (A) Estimated age-profiles for genetic risk for I10 "essential (primary) hypertension" (left) and I25.1 "atherosclerotic heart disease of native coronary artery" (right) fitted under the univariable (purple) and multivariable (green) approaches. For I10, the solid line indicates the posterior mean and the shaded area the 95% credible interval; For I25.1, the solid and dashed lines indicate the means for the two clusters of variants. Comparisons for all diseases are shown in S10 Fig. (B) Estimated incidence by age for K80.2 "Calculus of gallbladder without cholecystitis" (left) and C44.3 "Other and unspecified malignant neoplasm of skin and unspecified parts of face" (right). The red solid line indicates the rate estimated from the UK Biobank (see S1 Supplemental Methods) and the dotted blue line indicates the fitted incidence curve from the parametric model. The P value indicates the Goodness-of-Fit test. Curves for all diseases are shown in S11 Fig. (C) Comparison of inferred genetic effect sizes (red curve) and those implied by the frailty parameters estimated from incidence rate within the UK Biobank (blue dashed curve).

parameters. For example, the baseline hazard rate of K80.2 "calculus of gallbladder without cho-lecystitis" is estimated to increase proportional to age to the power of 1.9, but with substantial frailty (scale parameter = 1.87, goodness-of-fit P = 0.93; Fig 5B). In contrast, the baseline hazard rate of C44.3 "other and unspecified malignant neoplasm of skin of other and unspecified parts of face" is estimated to increase more rapidly with age (power of 3.58), but with lower frailty (scale parameter = 0.94; P = 0.76). It should be noted that the simple parametric model can be rejected at P < 0.01 for only one (J45.9, "other and unspecified asthma") of the 24 disorders, with the main discrepancy being a reduction in incidence among the eldest UK Biobank partici-pants compared to the fitted model, which may potentially be explained by selection bias in recruitment and competing risks of multi-morbidity. We note that the estimated magnitude of frailty is typically sufficient to lead to an elevated false positive rate of the test.

Previous work has demonstrated that the magnitude of the diluting impact of frailty on effect sizes in longitudinal models can be predicted using the incidence and frailty distribution parameters [17]; notably the implied effect size at a given age is reduced by a factor propor-tional to the prevalence at that age multiplied by the variance of frailty distribution; see Materi-als and Methods. We therefore compared inferred (univariable) curves for genetic variants against that implied by the fitted frailty model (Fig 5C). In 17 of the 24 diseases we find that while the estimated frailty predicts a decreasing genetic effect size with age, the observed decrease both starts earlier and is of a larger magnitude than expected (S12 and S13 Figs). Importantly, the estimated effect size tends to decrease substantially even when the prevalence of the disease is very low. We therefore conclude that, even after accounting for independent unmeasured factors that influence disease risk, genetic relative risk decreases with age.

## Discussion

Genetic factors influence lifetime risk for common and complex diseases through modulating a large number of molecular, cellular and tissue phenotypes, many of which are also likely to be affected by acute exposure and persistent environment [26–28]. Despite such complexity, remarkable progress has been made in identifying factors, both genetic and non-genetic, that influence risk, each of which may only have a small effect, but which, in aggregate, have sub-stantial and clinically relevant predictive value [29–31]. To date, while multiple studies have shown genetic interactions with contexts such as age, sex and environment when modulating disease risk [32,33], the extent to which polygenic risk prediction can be improved by allowing genetic risk to be modulated has not been fully explored. Here, we set out to measure how one specific aspect of individual context, namely age, can modulate genetic relative risk. For exam-ple, whether there are windows during which genetic risks are particularly relevant to disease and, conversely, other windows in which genetics plays a lesser role. The methods introduced here provide a flexible framework in which to address this question, as well as considering het-erogeneity among diseases and classes of variants.

By applying the methods to data from the UK Biobank, we have identified four aspects of the relationship between age and genetic relative risk. First, we have shown that for many dis-eases, but certainly not all, there is statistical support for a non-constant relationship between age and the influence of genetic risk. Second, in such cases, genetic risk has the greatest effect at earlier ages, though the magnitude and form of the drop-off varies among diseases. This result agrees with and generalises earlier reports [8,16]. Third, there is relatively little evidence for different groups of variants having substantially different relationships between age and risk; where we identify weak evidence for multiple classes, the differences are in terms of the magnitude of the downward slope. Fourth, the drop-off in relative risk with age cannot be ascribed to hidden variation in unmeasured risk factors. We note that the drop-off in impact

of genetic risk factors does not mean that they are not relevant in predicting later disease, which is typically when most diseases occur. Rather, our results imply that the factor by which genetic factors increase risk above baseline for someone in their 40s may be several fold higher than for an equivalent person in their late 70s (S7 Table). For example, the factor by which being in the highest decile of genetic risk for I25.1 "atherosclerotic heart disease of native coronary artery" increases incidence over baseline between 45 to 50 years old is 6.62, compared to only 2.4 between 70 to 75 years old. Put another way, by assuming a constant effect over age, we may both underestimate the absolute risk for young individuals with a high polygenic burden and overestimate the absolute risk for older individuals with a high polygenic burden.

What biological processes could lead to a diminished influence of genetic risk over time; in effect a decrease in heritability of a trait with age? Genetic risk factors, unlike environmental ones, are present from birth, while non-genetic risk factors tend to accumulate and evolve over time. Such a difference could lead to a reduced impact of genetics over time if genetic risk primarily influences developmental pathways, while non-genetic risk affects separate and later-impacting pathways, such as those involved in adult homeostasis (Fig 6A). However, there are several contexts where genetic and non-genetic risk are, at least in part, mediated by the same factor, such as the impact of LDL cholesterol on cardiovascular disease. In such cases, statistical interactions between genetic factors and the environment (or potentially among genetic risk factors) could have a diluting effect on genetic risk in a manner similar to frailty (Fig 6B). Here, an interaction means that the combined effects of the genetic and non-genetic risk factors is worse than expected from their independent contributions. Biologically, such an interaction could arise from threshold models of homeostasis (meaning the system can buffer only up to a certain level of challenge), though many other biological processes could potentially lead to statistical interactions at the population level. Risk process models such as threshold models [34,35] provide a potentially rich framework for modelling such effects, though which models are distinguishable from cohort-level data such as biobanks remains an open question.

We note that our work has multiple limitations, including the focus on a single cohort that is dominated by the single ethnic group, the lack of power for genetic discovery for many diseases of interest and the focus on the single context of age, rather than a much wider set of potential modulating contexts. Moreover, our work, while revealing an underlying pattern, has not advanced our understanding of the biological causes for it (or why some diseases show this but others do not). We hope, however, that the findings motivate future research into the biological causes and clinical consequences of such age-varying genetic relative risk. For example, identifying environmental exposures or comorbidities that are compatible with the decreasing risk framework could provide information on the biological mechanisms (for example, how genetic contributions to LDL at different ages impact risk for cardiovascular disease at different ages). Additionally, targeting additional recruitment for GWAS on both early and young age groups could inform more accurate estimation of age dependency in genetic relative risk, as could the collection of cross-generational longitudinal data.

Whatever the cause of age-varying genetic relative risk, our results have several implications for the use of genetic risk factors in the genetic analysis and prediction of disease risk. Most obviously, genetic risk prediction for disease must take age into account both in terms of its impact on disease incidence, but also in terms of its impact on genetic relative risk. We note, however, that even with diminished genetic relative risk, the clinical utility of polygenic risk prediction may still be higher among older individuals (for example, in terms of reclassifying individuals as above or below a threshold in absolute risk). For most of the diseases studied here, the inference of a single age-profile does mean that the rank order of genetic risk for an individual is stable over time. However, it implies that integrated prediction from genetic and non-genetic risk factors [36–38] will have to consider the evolving contribution of genetics

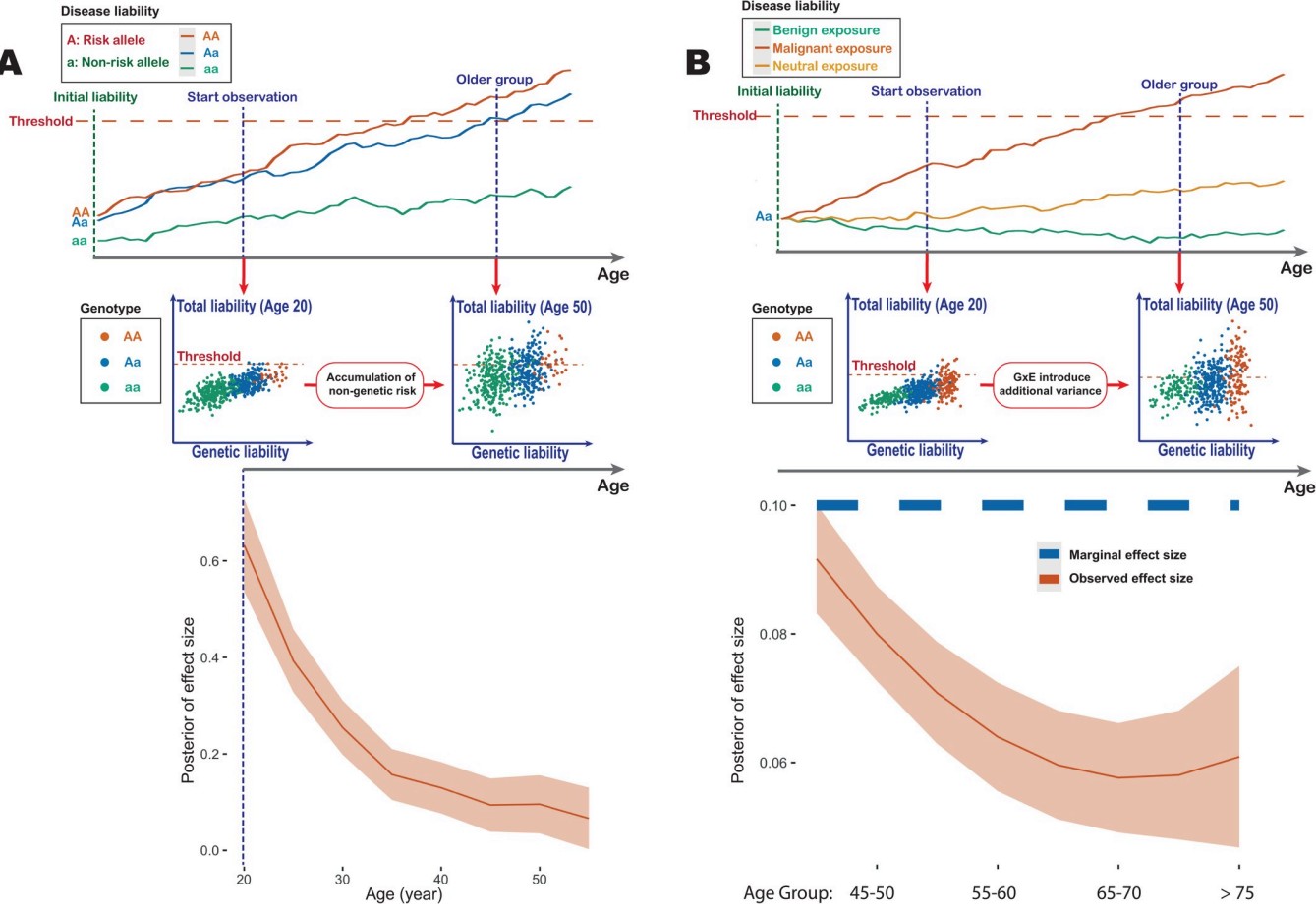

**Fig 6. Models for a decreasing influence of genetic risk with age.** (A) A threshold model, in which each individual has a disease "liability" which evolves over age. Disease onset occurs when liability crosses a threshold. The upper panel shows example trajectories, where genetic risk alters only the liability baseline. The middle panel is a schematic representation of a simulation in which genetic risk affects developmental pathways at birth, while non-genetic risk accumulates over time. The lower panel shows an estimation of the effect size from a simulated dataset of UK Biobank sample size (see S1 Supplemental Methods). (B) Interactions between genetic and environmental risk factors can create a distribution of effect sizes for a specific genotype. The upper panel shows example trajectories, where the environment influences the slope of the trajectory. The middle panel shows illustrative examples of the liability distributions among individuals at different ages. Those individuals at highest risk (with both the risk allele and risk environment) enter disease earlier, diluting the apparent effect size at a later age. The lower panel shows simulation results under such a model using realistic parameters from UK Biobank (see S1 Supplemental Methods).

over age. For diseases with multiple age profiles, even the rank order of genetic risk among individuals could change over time. Finally, because contexts beyond age, such as sex and environment, modulate genetic risk [16,36,39], each of these will induce its own age-specific profiles. As a consequence, effective genetic prediction will most likely be driven by empirical models that can benefit from access to large and well-measured populations, such as population-scale biobanks.

## Materials and methods

Full technical details are given in the S1 Supplemental Methods and S1 Analytical Note.

### Data preparation

We use the genotype data, basic demographic data and Hospital Episode Statistics (HES) data from 409,694 individuals of British Isles ancestry in the UK Biobank dataset [20]. 31 ICD-10

codes were identified with a prevalence above 5% and for which at least 20 independent associated variants were previously identified using the TreeWAS model [21]. Of these, we analysed 24 that correspond to specific disease conditions (as opposed to procedures). These are listed in Table 1. For each ICD-10 code, we combined the primary and secondary diagnoses from the HES data. We used the starting date of the first episode that records the disease diagnosis to define the age of disease onset, which is calculated as the difference between onset date to the month of birth (due to data privacy, we only have access to birth information specified to year and month). The onset age is rounded to whole years. For each ICD-10 code, only the first recorded diagnosis of each individual was used. For the population under observation, we also computed the age at observation endpoints, which is either the age of individuals at the last update of the data set (here 2018-02-14) or the age of death if a death event is recorded. We categorised the disease onset age into 8 age intervals, the first and last of which are "before 45 years old" and "after 75 years old", with 5-year intervals in between.

We then constructed interval censored data sets for the selected disease. Each age interval is an observation window of all healthy (alive and without onset of target disease) individuals who survived past the starting point of the interval. Onset of disease and exiting the study (death or no further records available) are recorded as "case" and "censored" events respectively. Events happening after this interval are considered right-censored at the end of the interval. We then performed case-control matching over the sub-population observed within each interval in two steps. First, we divided the sub-population into a disease group and a control group. The disease group are those who have disease onset within the interval, and the control set are those who do not, including individuals who have disease onset after the age interval, so long as they remain healthy before the endpoint of the interval. This is what the term "interval-censored" means. In survival analysis, the control groups are called "censored", and the age at the "censoring" event is also needed for unbiased estimation. If a censoring event is observed within the age interval (i.e. the age of a death record or the last update in the UK Biobank is before the age interval end point), we used the age at the censoring event. If an individual does not have an event record within the age interval, we take their age at the end of the interval, regardless of their future events. Second, for each case in the disease group, we pick four nearest neighbors (without replacement) from the control group, matching sex, BMI, year of birth and 40 genetic principal components. The covariates are available within the UK Biobank data set, over which we computed the principal components across the British Isles ancestry population. We then compute the Euclidean distance of the principal components to find the nearest neighbors in the population.

## Estimating age-dependency of genetic risk score in prediction

The SNPs of interest are obtained through prior TreeWas analysis, where we select variants that have Bayes factors (BF) $\geq 5$ (BF is computed for a single variant's effect over the TreeWas model) and posterior probability (PP) $\geq 0.99$ for target diseases [40]. We further filtered the set of SNPs to ensure LD-independence (loci kept with absolute Pearson correlation coefficient smaller than 0.2).

To assess whether the collective effect of risk variants, as captured by a combined genetic risk score (or polygenic risk score—PRS), showed profiles of age-varying risk, we used the case-control matching procedure described above with five-fold cross-validation, keeping 20% of case-control pairs for each age interval as test sets, and estimating effect sizes for the selected variants in the remaining 80% of case-control pairs using multivariable logistic regression (including age, sex and 40 genetic PCs). The effect of the PRS on risk within each age interval in the test set was then estimated (again with logistic regression and covariates). We estimated

the odds-ratio for the top decile of risk and the top 20% of risk versus the whole population, using 20 repeats of the procedure to obtain the bootstrap sampling distribution.

### Estimating age-specific effects of genetic risk factors

We used a standard GWAS approach to identify the risk and protective alleles at each locus [41], over the case-control matched dataset described above. The first 40 genetic principal components are taken as covariates. For all loci that have protective minor alleles (odds ratio < 1), we switched allele labels to assign consistency of risk direction.

To obtain an unbiased estimate of genetic risk effect size over age, we used a proportional hazard (PH) model to estimate the genetic hazard ratio for different age groups, using the case-control matched data set. Within each of the 8 age intervals, we applied the PH model to the disease group and control group, accounting the censoring effect. We used a univariable model to estimate the effect size of each variant separately and a multivariable model to estimate effect sizes for all variants jointly. Covariates include the first 40 genetics PCs of the UK Biobank and are regressed out for each interval. Both the point estimate and standard error of effect sizes are obtained for each variant within each age interval. These summary statistics are used subsequently for curve-cluster fitting.

### Bayesian clustering of genetic risk profiles

To group variants that have similar age-dependency, we applied a Bayesian clustering of curves model described (see S1 Analytical Note). The model assumes each variant has an age-dependent effect profile which is generated from a mixture of curves model. The mean and standard error of the age-specific effects for individual variants described above are the inputs of the model, from which we infer the underlying generative latent curve. The model allows vertical translation in the generative process (i.e. the likelihood won't change much if the profile of the variant is far from the latent profile, as long as the shape of the variant curve is similar). The latent curve is a spline whose smoothness is controlled by changing the degrees of freedom. For detailed specification of model and hyper-parameters, see the S1 Analytical Note. Inference is performed by an EM algorithm and was repeated 20 times with random initialization of variables (see S1 Analytical Note). The highest likelihood sequence was retained. Since EM only provides a point estimate, we estimate the curve's credible interval using a variational approach. The inferred profiles with 95% credible intervals are shown in S4 Fig. The derivation and proof of the approach are provided in S1 Analytical Note.

### Permutation testing for genetic effect heterogeneity over age

To provide robustness in testing for age heterogeneity, we carried out a permutation test, using the likelihood ratio test statistic (for fitting a non-uniform genetic risk over age) for both the original data set and permutation samples to obtain the permutation p-value. The likelihood ratio test compares an alternative model with linear and quadratic genetic risk over age and a null model assuming a constant effect over age (see S1 Supplemental Methods and S1 Analytical Note).

To perform permutation tests, we kept the matched case-control structure and then sampled case-control pairs for each age interval, while fixing the onset age distribution for permutation samples. We repeated the procedure 10,000 times to obtain permutation samples, and computed the likelihood ratio for each sample. We note that the likelihood ratio does not include the prior term for spline coefficients, while EM finds the Maximum a Posteriori (MAP) estimate (see S1 Analytical Note), which will give a likelihood slightly lower than the MLE estimate. Under the permutation test framework, the p-values will be consistent as long

as the same test statistics are used for both the original data set and permutation samples [42]. We further checked that the difference between MAP estimation and MLE estimation is negligible.

The EM procedure (see S1 Analytical Note) is initialised randomly 20 times for the observed data and each permutation sample to compute the likelihood ratios. The permutation p-value for each disease is obtained from the likelihood ratios. We correct for multiple testing using FDR, with the corrected q-values shown in Table 1 (when a multivariable approach is used to estimate effect size) and S1 Table (when a univariable approach is used).

In order to determine the optimal number of clusters for each disease, we performed permutation tests using the same procedure, considering the addition of each new cluster. For adding the k+1 cluster, the alternative model has k+1 clusters and the null model has k clusters. All models are fitted with quadratic polynomials (see S1 Analytical Note). Again, we computed the likelihood ratio statistics for both the observed data set and permutation samples to obtain p-values. This analysis is performed over all diseases and adjusted for multiple testing with FDR. We note that we found no compelling evidence supporting a model of more than two clusters for any disease. The p-values and q-values for the test of two clusters are shown in Table 1 (when effects for individual variants are estimated jointly) and S1 Table (when effects for individual variants are estimated using a univariable model).

## Estimating effects of unobserved risk background

To estimate the effect of unobserved risk factors, we assumed an individual hazard model that has a frailty coefficient and baseline hazard. The frailty coefficient has mean 1 and a scale parameter that controls the variance of population hazard rate. We chose the baseline hazard to be a power function of age. We fitted the model to the empirical incidence rates in the UK Biobank. The empirical incidence rate at a specific age is computed as the number of individuals who have first onset of the target disease within this age year, divided by the number of healthy individuals at risk at the beginning of this age year. We then fitted the parametric hazard to the empirical incidence rate until age 70, and finally subtracted the intercept from the empirical incidence rate to match the parametric form of the hazard rate. We fitted the model by minimizing square error using the Nelder-Mead method. The fitted incidence curves are compared with empirical curves for all diseases (S11 Fig). We also computed a Goodness-of-fit p-value for each disease, comparing the match between fitted and empirical three-year incidence rates using a Chi-square test statistic. The Goodness-of-fit p-values are shown in S6 Table. We used the inferred parameters to predict how genetic effects are expected to be diluted by the presence of frailty (S13 Fig; for technical details, see S1 Supplemental Methods and S1 Analytical Note).

## Simulation

In our simulation, we generate a risk profile over age for each variant from underlying curves with different slopes. The individual risk is then computed at different ages, which are then used to generate disease incidence events over the simulated population. We choose the population size to be 50,000, which is comparable to our empirical case-matching population size (the set of common diseases analysed each have ~10,000 cases in the UK Biobank and we match each case with four controls). We simulated 50 SNPs (MAF of each SNPs are sampled from uniform distribution 0–0.5). The risk effect for each SNP is sampled from a profile which changes linearly with age. The individual hazard within a specific age interval is computed as the exponential of genetic risk multiplied by a linearly increasing baseline hazard ratio.

For each interval, we simulated the time to the next event using a homogeneous Poisson process with the defined individual hazard rate. An individual with no event in this interval is considered as observed (censored). We record only the first event as the onset of the disease. The simulation is performed over a 40 year duration divided into eight 5-year intervals, as most of the disease onset occurs between ages 40–80 years old in the UK Biobank. In order to represent the end of observation (study drop-out) or death events in the cohort, a competing censoring process is sampled using a Poisson process of constant rate. The dropout/death and disease onset events are combined and we keep the first event, labelling it as either disease or censoring. For parameters setting in the simulation, see S1 Supplemental Methods.

To test our statistical model for inferring age-varying genetic effects, we simulated a population using the scheme described above and analysed it using the methods described above to infer the genetic risk profiles over age and the underlying curves that generated them. We simulated the cohort with different values of the slope, which represent different age dependencies, and tested whether our method could recover the simulated values. We then assessed the power of the statistical test to detect age-varying genetic effects. We simulated the genetic risk profile with the slope ranging from -0.01 (linearly decreasing with age) to 0.01 (linearly increasing with age), with a step size of 0.0001. The simulated population is analysed using the null model of a constant effect with age, and an alternative model of either a linear model, or a quadratic polynomial curve. A likelihood ratio test is performed to calculate the p-value, and we calculate the power of rejecting the null at a threshold of $p = 0.05$. For each slope, the simulation was repeated for 400 times to estimate the power and its standard error.

To test our statistical model for detecting multiple clusters of genetic risk profiles, we simulated disease cohorts with five (10%) of the variants that had effect sizes generated from a non-constant latent profile, while the effect sizes for the remaining 45 variants had a constant (age-invariant) effect. We assessed our model as to whether it can detect the presence of multiple clusters. The simulated cohorts are analysed with both a null model of a single quadratic polynomial curve, as well as the alternative model of two quadratic polynomial curves. For each simulation, we compute the p-value for the likelihood ratio test comparing two clusters against one cluster, measuring power at $p = 0.05$. We varied the slope of the non-constant profile to test how different the curve needs to be from a constant effect to be distinguishable by our model. Power is computed for slopes ranging between -0.0375 and 0.0375, with a step size of 0.00025. For each slope, the simulation was repeated for 400 times to estimate the power and its standard error.

To model possible mechanisms for the observed decline in genetic risk with age we simulated a threshold model in which each individual has an unknown "liability", which evolved over time [43]. For a specific disease, onset occurs when an individual's "liability" passes a certain threshold. We simulated a liability model for 50,000 individuals with a single genetic effect that alters the starting point of liability. Genotypes were simulated with a risk allele frequency 0.3. The liability is simulated as a stochastic process with starting points altered by genotypes. We then simulated increments of liability from a Gaussian distribution which controls the drift and variance of the stochastic process. The stochastic process models the disease risk increase over age through the drift, and the correlation of increments induced by the variance of Gaussian distribution creates a "momentum" such that an individual's health status tends to improve or deteriorate over years at a similar rate. We simulated for 60 years and considered an individual to have an onset of a disease when the liability (arbitrarily) reaches 0. We then estimated the effect size of the risk allele over the age interval 21–60. For parameters setting in simulating the stochastic processes, see S1 Supplemental Methods.

To consider whether the decreasing pattern could be explained by interactions (either gene-by-environment or gene-by-gene) we performed additional simulations. We modelled the

interaction of a focal genetic effect with other unobserved risk factors. Assuming the effect size interacts with environmental or other genetic factors, the effect size for each individual is generated from a positively defined distribution. We can show that the estimated marginal effect size will be increasingly underestimated as age increases for all positive defined probability distributions (see S1 Analytical Note). We then performed a simulation using the parameter settings described at the beginning of this section, but sampled an effect size for each individual from a gamma distribution. The effect size for each individual remains constant over age intervals. We then inferred the posterior of effect size, presented in Fig 6B. We note that this model is a generalisation of the concept of frailty in which one allele has greater frailty than the other.

## Supporting information

**S1 Supplemental Methods. Technical details.**
(PDF)

**S1 Analytical Note. Analytical details for models.**
(PDF)

**S1 Fig. The prediction power for combined genetic risk scores for additional diseases.** In each plot the odds ratios for the 80th (blue) and 90th percentiles of a combined genetic risk score within matched case-control samples (four controls for each case) are shown for each age interval; points indicate the average odds ratio of twenty five-fold cross-validation analyses with lines indicating the 95% confidence interval.
(PNG)

**S2 Fig. Simulation studies.** (A) A simulation with frailty showing that the inferred effect (red) deviates from the underlying effect size (blue dashed line). The variance of frailty in this case is 0.82. (B) Effect of frailty on the false positive rate. The x-axis shows the variance of the frailty distribution, with a larger variance indicating stronger frailty, while the y-axis is the false positive rate of rejecting the true model of constant effect over age. The inferred curve does not deviate from uniformity when the frailty variance is smaller than 0.1. (C) Coverage analysis. The blue curve shows the probability that 95% posterior credible interval covers the true genetic profile and the shaded area is the 95% confidence interval of the coverage estimate. (D) Simulation to test the impact of selecting healthier individuals of older age. Selection bias towards healthier older people is simulated by changing the baseline hazard over age, such that a negative slope indicates a population in which older people are biased away from having disease. The blue solid line shows the false positive rate of rejecting the null hypothesis of uniformity for a baseline hazard with different slopes; the shaded area shows the 95% confidence interval. Genetic profile estimation uses a quadratic polynomial throughout.
(PNG)

**S3 Fig. Analysis of empirical data with curves of increasing complexity.** A likelihood ratio test is performed against a constant effect model (DF = 1) over age, for models with different smoothness. Smoothness is controlled by the degree of freedom of the spline basis, where we tested linear (DF = 2, blue), quadratic polynomial (DF = 3, red), cubic polynomial (DF = 4, orange), spline with one knot (DF = 5, green) and spline with two knots (DF = 6, grey). The red dotted line indicates P = 0.05.
(PNG)

**S4 Fig. Posterior estimation of genetic risk over age for all diseases.** The solid red curve indicates the posterior mean, and the shaded region is the 95% credible interval.
(PNG)

**S5 Fig. Posterior curve estimates when two latent profiles are fitted for six diseases with moderate evidence of multiple profiles (P < 0.1).** The red and blue curves with corresponding shades show the profile means and 95% credible intervals for each profile.
(PNG)

**S6 Fig. Age-varying disease risk profiles for different self-identified races or ethnicities.** (A) Inferred cluster profiles for the six disorders presented in Fig 4 using all subjects in UK Biobank regardless of ethnicity. The blue and red curves in the last two figures indicate the means for the two clusters of variants, where there is evidence for multiple profiles within the British Isle ancestry group. Curves for all diseases are shown in S7 Fig. (B) Inferred cluster profiles for individuals self-identified as Black ethnicity in the UK Biobank. "Gastritis and duodenitis" (ICD-10 code K29), "low back pain" (M54.5), "Arthrosis" (M19.9) and "Asthma" (J45.9, P = 0.006). We also include the permutation P-value for asthma as its age profile has a uniquely increasing risk profile. (C) Inferred cluster profiles for individuals self-identified as South Asian ethnicity; "primary (essential) hypertension" (I10) and "Gastritis and duodenitis" (K29). The solid line indicates the posterior mean and the shaded area the 95% credible interval; Numbers in boxes indicate the number of variants in each cluster; All estimates are made with quadratic models for age-varying risk profiles.
(PNG)

**S7 Fig. Posterior estimation of genetic risk over age using all ancestries in UK Biobank for all diseases.** The solid red curve indicates the posterior mean, and the shaded region is the 95% credible interval.
(PNG)

**S8 Fig. Risk profiles using external variant sets and disease phenotype definitions.** (A) Inferred cluster profiles using associations collected from GWAS Catalog for two disorders: "primary (essential) hypertension" (I10) and "antherosclerotic hearth disease" (I25.1). (B) Inferred cluster profiles for two phecodes: "172.20" (other non-epithelial cancer of skin, ICD-10 codes: C44.3 and C44.5) and "411.20" (myocardial infarction, ICD-10 codes: I21.9 & I25.2). The solid line indicates the posterior mean and the shaded area the 95% credible interval; Numbers in boxes indicate the number of variants in each cluster; All estimates are made with quadratic models for age-varying risk profiles.
(PNG)

**S9 Fig. Comparison of receiver-operator curves (ROC) under different age assignments for training and testing sets.** The population is divided into a younger group and an older group, where GRS and ROC are estimated from one of the groups. Each plot shows the ROC for four combinations: GRS and ROC are both computed from younger (Young-young; blue) or older group (Old-old; red); GRS is computed from the younger group and ROC is computed from the older group (Young-old; green) and the other way around (Old-young; orange). The area under the curve (AUC) metrics are shown in the top left. C44.5 "other and unspecified malignant neoplasm of skin of trunk", M06.9 "rheumatoid arthritis, unspecified", E10.9 "type 1 diabetes mellitus without complications", K80.2 "calculus of gallbladder without cholecystitis", and I25.1 "atherosclerotic heart disease of native coronary artery" are ICD-10 codes that have the biggest AUC differences between the blue and red ROCs. M54.5 "low back pain" is the only disease that has a larger AUC in red ROC than blue ROC. Regardless of whether the GRS and ROCs are computed from the same age group, we use 80% of the sample to compute the GRS and 20% for the ROC to match the sample sizes under each condition.
(PNG)

**S10 Fig. Comparison of effect size estimation using multivariable and univariable methods for all diseases studied.** A quadratic polynomial model is fitted to the estimated effect size in both cases, which is shown as two curves: green (multivariable) and purple (univariable). (PNG)

**S11 Fig. Estimated incidence by age in the UK Biobank for all diseases studied here.** The red solid line indicates the rate estimated from the UK Biobank (see S1 Supplemental Methods) and the dotted blue line indicates the fitted incidence curve from the parametric model. (PNG)

**S12 Fig. Comparison of inferred genetic risk profiles and those predicted from fitted frailty models.** A) A likelihood ratio test of deviation from the fitted frailty model (red), compared with the likelihood ratio test of deviation from a constant effect model. Four diseases have $Q < 0.05$ after correcting for multiple testing. All inferences are performed on the univariable estimation of variant effect size because the fitted frailty should include both genetic and non-genetic factors. B) Paired t-test of the gradient of frailty and our inferred curve, identifying 17 out of the 24 diseases analysed where the inferred genetic risk profile slope is steeper than that implied by the inferred frailty parameters. (PNG)

**S13 Fig. Comparison of genetic risk estimated using quadratic polynomials and that predicted by a frailty model.** Comparison of fitted latent curves (red curve for the mean and shaded region for the 95% credible interval, estimated using the univariable approach) and latent profiles implied by the fitted frailty effect (blue dashed line), for all 24 diseases analysed here. (PNG)

**S1 Table. Summary of evidence for age-varying genetic risk when fitted with univariate model.** Summary of ICD-10 disease codes analysed and evidence for age-varying effect sizes and number of age-profile classes, fitted with a univariable model and quadratic polynomial. Details are for Table 1. (PNG)

**S2 Table. Summary of odds ratios for the 80th and 90th percentiles of genetic risk score within each age interval.** (A) Mean odds ratio of the 90th percentile GRS over the population average. (B) Mean odds ratios of the 80th percentile GRS over the population average. Parentheses contain the 95% confidence intervals of the mean. (PNG)

**S3 Table. Summary of changes in genetic risk contributions.** Summary of changes in genetic risk contributions from before 45 years old to after 75 years old, when risk profiles are fitted using a linear model. (PNG)

**S4 Table. Posterior mean risk profiles for all diseases.** Posterior mean risk profiles for all diseases analysed here, fitted with a single quadratic polynomial. Standard errors are also provided. Values are the mean effect size within the age interval for individual variants. (PNG)

**S5 Table. Comparison of area under the curve (AUC) metrics under different age assignments for estimation and prediction for all diseases.** The population is divided into a younger group and an older group, where GRS is estimated in the training set from one group and

receiver-operating curves (ROCs) are computed from the testing set in one group. By the combining age assignments for the training and testing sets there are four conditions: GRS and ROC are both computed from the younger (third column) or older group (fourth column); GRS is computed from the younger group and ROC is computed using the older group (fifth column) and the other way around (sixth column). Regardless of whether the training data and testing data are from the same age group, we use 80% of the sample as the training set and 20% as the testing set to match the sample sizes of each condition.
(PNG)

**S6 Table. Estimated parameters of frailty from the UK Biobank.** The fitted model has a hazard rate of $h_i = u_i \, \gamma t^k$, where $u_i \sim$ Gamma(shape = $1/\theta$, scale = $\theta$).
(PNG)

**S7 Table. Comparison of risk between an early age group and a late age group.** Comparison of baseline hazard rate (population-level risk), genetic risk factor effect size and absolute hazard rate (baseline hazard multiplied by genetic risk factor) for an early age group and a late age group across the diseases studied here. The genetic risk factor and absolute hazard are computed from the group with the highest decile of genetic risk.
(PNG)

## Acknowledgments

This research has been conducted using the UK Biobank Resource; application number 12788.

This work uses data provided by patients and collected by the NHS as part of their care and Support. Computation used the Oxford Biomedical Research Computing (BMRC) facility, a joint development between the Wellcome Centre for Human Genetics and the Big Data Institute supported by Health Data Research UK and the NIHR Oxford Biomedical Research Centre. The views expressed are those of the authors and not necessarily those of the NHS, the NIHR or the Department of Health. We thank Brieuc Lehmann and Luke Jostins-Dean for discussion and comments on the manuscript.

## Author Contributions

**Conceptualization:** Chris Holmes, Gil McVean.

**Data curation:** Xilin Jiang.

**Formal analysis:** Xilin Jiang, Chris Holmes, Gil McVean.

**Funding acquisition:** Chris Holmes, Gil McVean.

**Investigation:** Xilin Jiang, Gil McVean.

**Methodology:** Xilin Jiang, Chris Holmes, Gil McVean.

**Project administration:** Gil McVean.

**Resources:** Gil McVean.

**Software:** Xilin Jiang.

**Supervision:** Chris Holmes, Gil McVean.

**Validation:** Xilin Jiang, Gil McVean.

**Visualization:** Xilin Jiang, Gil McVean.

**Writing – original draft:** Xilin Jiang, Chris Holmes, Gil McVean.

**Writing – review & editing:** Xilin Jiang, Chris Holmes, Gil McVean.

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
