## [Decision Letter · Decision Letter 0]

29 Mar 2021

Dear Gil,

Thank you very much for submitting your Research Article entitled 'The impact of age on genetic risk for common diseases' to PLOS Genetics.  Please accept our apologies for the delay in returning this review to you. It was extraordinarily difficult to secure reviewers, and their responses were delayed, though we were fortunate in securing three highly qualified reviewers for your work.

Based on the reviews, we will not be able to accept this version of the manuscript, but we would be willing to review a much-revised version. We cannot, of course, promise publication at that time.  Your revisions should address the specific points made by each reviewer. Please also provide a detailed list of your responses to the review comments and a description of the changes you have made in the manuscript.

In addition to addressing comments of the reviewers, please explain and justify the limitation of analyses to persons of British Isles ancestry. Please provide parallel analyses in supplementary materials that include all UK Biobank subjects regardless of ancestry, with any desired adjustments explained and justified, and describe them briefly in the main body of the paper along with description and discussion of similarities and differences from the British Isles subgroup analysis. Stratified analyses by ancestral group may also be included and commented upon at your discretion.

Please also add a limitations section to the Discussion.

If you decide to revise the manuscript for further consideration at PLOS Genetics, please aim to resubmit within the next 60 days, unless it will take extra time to address the concerns of the reviewers, in which case we would appreciate an expected resubmission date by email to plosgenetics@plos.org.

[LINK]

We are sorry that we cannot be more positive about your manuscript at this stage. Please do not hesitate to contact us if you have any concerns or questions.

Yours sincerely,

Teri Manolio

Guest Editor

PLOS Genetics

David Balding

Section Editor: Methods

PLOS Genetics

Reviewer's Responses to Questions

**Comments to the Authors:**

Reviewer #1: Jiang et al detail a new method to determine the non-linear relationship between age and genetic effects, using the UK Biobank as the applied dataset. The systematic identification of unique genetic profiles that differ by age is a meaningful contribution to the literature, as is the elegant exploration of different factors (frailty, bias) that could influence these relationships. The manuscript as it is currently framed is somewhat in between the presentation of a new method and the application of the new method to better understand genetic risk in the UKBB. It would be beneficial to the manuscript to do a little bit of a tweaking to help the reader understand how this method could be used in their own workflows. This would require showing how the identification of these profiles is important when estimate genetic risk. While this would require additional analyses, I think it would make the manuscript more impactful. In short, when I read this paper I’d like to have a figure to point to and say “Ah! If I hadn’t used this method to find non-linear relationships by age, I would have created a model and given back false risk estimates to my patients!” A few major and minor points are outlined below.

Major points.

• The authors estimate age-specific effects for each SNP within each interval. However, the original set of variants were selected according to a previously published TreeWAS approach, which was conducted across the entire UK biobank sample. Is this original selection weighted towards younger individuals, and therefore younger cases? This may be outside of the scope of this manuscript, but it would be interesting to know how much of the decrease in genetic risk is due to selection bias from this original analysis (i.e. do different loci contribute to risk as we age?)

• Throughout the manuscript it would be beneficial to include the actual numbers, instead of referring to trends, especially when the numerical results are in the supplemental tables (or require going through another dense table). For example, page 8 lines 120-123 include phrases such as “dramatic decrease” and “gradual decline”. It would help to include some numbers to help contextualize these phrases.

• Figure 2 is very helpful in understanding the method, as well as the framework of the manuscript. It would be even better to include a panel within Figure 2 that refers to the “Age-profiles for genetic risk scores” to contrast the different approaches. (Then switch Figure 2 to Figure 1 and push currently Figure 1 down.)

• The new method detailed is to determine non-linear relationships between age and genetic risk. However, the majority of the results sections do not fully explore the non-linear changes in risk. For example, in the “Application to common diseases in the UK Biobank” section (pages 10-11), the authors compare risk at 45 and 75. However, there are some results in which risk is largely unchanged up until a certain age (Figure 4F), which would support a non-linear relationship and warrant expansion. (I basically am saying that you are underselling your method and findings. It would be great to have more on these non-linear trends!)

• The authors stop one step short of their stated motivation, which is to see how risk prediction can be improved by allowing factors to be modulated by context. It would be great to include this in the manuscript. This method clearly shows how you can identify risk profiles that change over age in a non-linear fashion. You could compare a standard PRS approach, in which age is added into the model with one set of effect weights for the PRS, to an approach in which effect weights are changed based on age bucket. I imagine the model fit would improve, especially for older populations, for those with non-linear trends with this approach compared to the more standard adding of age as an independent linear risk factor. In short, going one step further and showing the consequences of ignoring these relationships versus addressing them when conveying an individual’s risk (ranking) would be important.

• It would be helpful to provide some discussion about specific UKBB results, such as why would there be nonlinear relationships between certain ICD-10 codes, genetic risk, and age? Is there any literature about these specific outcomes that would explain some of this? Biological plausibility would add another layer of impact.

Minor points.

• It’s unclear why the “Data preparation” section is included in the Results. It seems redundant with the Methods section. This paragraph would serve better as a way of introducing the age distribution of your sample, such as how many people fit into the different intervals for each outcome and if they are unbalanced in other ways (i.e. your typical Table 1 for epidemiological studies).

• The second ICD-10 in Table 1 technically has a prevalence of 0.48%, which is lower than the 0.5% in the methods section (and first results section paragraph).

• Please include confidence intervals for the odds ratio estimates presented throughout the manuscript. (For example, page 8 line119)

• Page 12, lines 216-217 say that there is a FDR>5%. Please put actual FDR (how much higher than 5%).

• Typo in Figure 2, panel C under “Estimate genetic risk profiles for all associated SNPs of a specific(e) disease”.

Reviewer #2: This paper uses UK Biobank data to systematically assess whether hazard ratios for disease-associated SNPs and polygenic risk scores vary by age. A number of papers have already shown that polygenic risk scores have age-dependent odds/hazard ratios or linear regression slopes for particular diseases and traits: age-specific effects have been observed for breast cancer [Mavaddat PMID:30554720 and PMID:25855707], prostate cancer [Conti PMID:33398198], breast and cervical cancer [Kachuri PMID:33247094], weight [Khera PMID:31002795], BMI [Mostafavi PMID:31999256], et cetera. This paper makes a contribution to that literature by focusing on age interactions, examining them in a systematic way across 24 common diseases in a large population-based cohort, and using appropriate methods for time-to-event data.

Weaknesses of the paper include:

--imprecise language ("biased"/"unbiased" without clarity as to the target parameter to be estimated; references to "effect" without specifying the relevant effect measure [e.g. age-specific hazard ratio conditional on the unobserved frailty vs. age-specific hazard ratio marginal over the unobserved frailty vs. age-specific risk difference marginal over the unobserved frailty);

--lack of clarity in the description of the methods;

--and a tendency to overstate the implications of these findings both in terms of prediction (which will depend on epidemiologic and clinical context) and in terms of underlying etiology (finding that a model with age-constant genetic hazard ratios or age-dependent hazard ratios or an accelerated failure time model or a liability threshold model is a better empirical fit does not necessarily imply that a gene "acts biologically" at one stage of life or another: e.g. under a "two-hit" process for carcinogenesis, I can come up with two plausible mechanisms that would lead to increasing hazard ratios with age, one where genes only act early in life, and one where they only act later).  

Specific comments follow.

Line 60-61: "... though see [6]." Spell out the specific reservations described in reference [6].

Line 62: "... has received relatively little attention..." Hm. Not how I would characterize the part of the elephant I'm familiar with. I do think it's fair to say the extent and impact of age-specific effects has not been thoroughly or systematically explored.

Line 63-63: "One aspect... is the role of age in modulating effects. Several studies have identified variants that influence age-at-onset... Similarly individuals with high PRS tend to have earlier age-at-onset..." The first sentence in this paragraph and the next 1.5 sentences are apples and oranges. Under a simple age-constant genetic hazard ratio model with no unmeasured risk factors, genotype/PRS will be associated with earlier age at onset. Association with age at onset in itself need not imply age modulates genetic effects. The clause "... genetic analyses of quantitative traits... whose effect size changes with age" is more relevant to the paragraph's topic sentence. And see the examples above of disease traits where age-specific hazard/odds ratios have been observed. Also pathogenic variants BRCA1/2 have well-known and large gradients in relative risks as a function of age.

Line 74: "...hidden risk factors can act to apparently reduce risk over time..." Not risk. Relative risk.

Lines 74-75: "... a phenomenon typically referred to as frailty in the epidemiologic literature." Hm. I don't know that confounding between measured and unmeasured risk factors induced by survivor bias is referred to as frailty--frailty refers to a set of models accounting for unmeasured risk factors. Anyway, safe to cut this clause. You introduce frailty and how it can lead to differences in age-specific hazard ratios (marginal over frailty) later.

Lines 90-94: I like the analyses that quantify the anticipated drop off in the age-specific marginal hazard rates due to unmeasured risk factors. But the last sentence is too strong. First, it's possible some frailty model that you did not explore could account for some of the effects. Second, and more importantly, it's really hard to argue from relative goodness of fit to different statistical models for "interaction" to underlying biological mechanisms (Siemiatycki PMID:7327838, Thompson PMID:1999681). E.g. a biological model where genes only act during puberty to set ones later susceptibility to breast or prostate cancer would be consistent with an age-constant relative risk model, as would a biological model where the genes are acting constantly across ages. Just tone this sentence down a little, and be more specific--e.g. "Our observations suggest that genetic relative risks (conditional on all other risk factors) vary with age" is on more solid ground than trying to tie that statistical phenomenon to an underlying biologic process. Especially when talking about so many different diseases, with distinct etiologies.

Line 106: "We used eight age intervals..." This seems out of place here. Plus not accurate, right? <45 and >75 are not five-year intervals. Can probably just cut this sentence.

Figure 1 and throughout: The authors never specify  what they mean by odds/hazard ratios for the 80th or 90th %iles. I assume that means top 20% resp. 10% versus the bottom 80% resp. 90%. If so, those are hard measures to interpret--they are driven in no small part by the folks in the lower (protective) %iles. Clinically and intuitively, the risk relative to population average is of most interest. Given this, and given the somewhat distracting duplication of every curve (80% and 90%), I suggest presenting the odds ratio/hazards ratio per standard deviation of the PRS. That also makes these results more directly comparable to other results in the literature, which report per-sd effects.

Lines 129-134, Methods, Supplementary methods, et cetera. I don't understand what was done here. Are the authors (a) creating case-control sets from all individuals at risk in the age interval (i.e. excluding those how have previously had an event or been lost to follow up before the start of the interval), and then performing logistic regression? Unconditional? Conditional on matching factors? Or are they (b) performing Cox proportional hazards regressions separately for each age interval (as in the Anderson-Gill model, allowing for age-interval-specific genetic hazard ratios)? Note: these two approaches differ in how they handle age-at-event within age intervals. In a Cox regression, as case who has an age later than the index case but still in the same interval would be treated as a control for the index case. In a logistic regression, none of the cases that occur in the interval would be treated as a control. For most diseases and most age intervals (the five year intervals), the two approaches shouldn't differ much. But for more common diseases and for the longer age intervals (<45 and >75), they might.

Other Qs: Why sample controls? Why not use the full risk set? Is this a computational issue or a way of getting fine-scale control of matching factors? If analysis (b) was performed, were the data restricted to cases and sampled controls or was the whole risk set used? If only the sampled controls were used, was the analysis run stratified by matched set (equivalent to conditional logistic regression)?

Line 138: "univariate or multivariate" I have no idea what this means. Perhaps univariable or multivariable, referring to regressions that do not adjust for measured factors (univariable) or those that do (multivariable)? Marginal (one SNP at a time, averaged over/ignoring the other SNPs) or multivariable (one model with alls SNPs included, a mutually-adjusted model)?

Line 152: Multivariate?

Line 206: ".... centered on effect size." You can safely cut this, and you should, as it is confusing without writing down what you mean in maths with specifically defined terms. "Effect size" is too vague. (The frailty is centered on the hazard conditional on observed risk factors).

Figure 5a. What's going on here? Are there four curves being plotted (see esp. second panel)? Red and green, dashed and solid? Uni/multi x empirical not assuming frailty model/assuming frailty model?

Lines 221-223: "... results were essentially identical under the two approaches..." What two approaches? "Univariate" versus "multivariate"? If so, why should it follow that "genetically-arising" frailty cannot explain the pattern? Again, I am confused as to what "univariate" and "multivariate" mean. If it means "marginal (one SNP at a time, averaged over/ignoring the other SNPs) or multivariable (one model with alls SNPs included, a mutually-adjusted model)," then the conclusion does not follow--I think the main point is that there might be other SNPs besides those included in the risk model that could be driving risk; different ways of modeling the included SNPs won't get at that.

Lines 252-253: I think you're on pretty firm ground with the last sentence, but suggest replacing "genetic risk" with "genetic relative risk."

Lines 260-262. Again, I would not say "little attention" has been paid to gene x age/sex/diet/behavior/environment interactions--generally and specifically in the context of risk prediction. Workshops have been held, RFAs have been issued, risk models have considered the inclusion of interaction terms and found some age interactions (although the boost in utility from including such interactions need not  be clinically meaningful). A few PMIDs you might find interesting: 24123198 27061572 28978193 28978192 28978190 21284036 33247094 32359158 25380502 27228256 31875139 29458155 22633398. Also, from this journal, PMID:19584936. And I didn't even pubmed Type 2 Diabetes or Coronary Heart Disease.

Lines 264-266: Best to be specific here. You set out to see if there were differences in genetic relative risks/odds ratios/hazard ratios by age. This is not the same as evaluating the "relevance" of genetic risk information in particular ages, either in terms of prediction (primary and secondary prevention) or in terms of etiology. For example, the per-PRS-standard deviation in genetic relative risk might be 3 for under 45s and only 1.5 for the 60-70s--but if disease incidence before age 45 is extremely rare, the benefits of PRS-targeted interventions might not outweigh the risks, whereas for older ages they might, due to the higher incidence. See also comments above about the challenges inferring biologically meaningful windows from age-specific relative risks.

Lines 274-275: "approaches that do not address the selection biases inherent in stratified analysis of longitudinal data." First, only one of these three cited references includes any data on age-specific genetic effects (and that only for one trait). Second, biased relative to what? In the prediction context, I'd argue we're estimating what we want when we estimate the age-specific relative risks marginal over the frailty. After all, we never get to measure the unobserved frailty! This should be noted--good news: if you care about prediction previous results are not useless and not necessarily "biased." If on the other hand you want to estimate the genetic relative risk conditional on the frailty, because you want to use that to make some inferences about underlying etiologic mechanisms, then yeah, the marginal relative risks are biased, and what looks like an age dependent effect may not be "real."

Line 284: "exponentially"--meaningless: everything is tautologically exponentially higher when your are comparing two groups (here 40s to 70s). The ratio is some positive number (other than 1) raised to some power: ergo exponential. Quote a range of observed ratios here (relative risk in 40s over relative risk in  70s from 1.5 to 6 or whatever it is).

Figure 6: What's going on in 6b? It's not clear to me how GxE interaction leads to different colored bits in the histogram. Are you saying 50% of the population is exposed, and they're the ones in the darker color? But how can that be? A mixture of normals is not ever normal, is it? Certainly not in general. Or it's a sum of G and product interaction G*E terms--in which case you should see G and GE contributions over the whole range. Methods lines 524-528 do not clarify this any. Supplementary methods 3.2.2. only gives methods for Figure 6.a. not 6.b.

Line 33: The liability threshold model is not any more generalized than the proportional hazards model. The liability threshold model is a specific parametric survival model  that implies a specific age-dependent genetic relative hazard function. It may make intuitive sense--perhaps too intuitive--and may provide a better empirical fit--although it's not as convenient to fit)--but I wouldn't call it any more generalized or richer than a proportional hazards model--especially as it is relatively straightforward to modify the Cox partial likelihood to allow for age-dependent relative hazards, as the authors have done here. 

Line 329: What are the parameters on the age and sex distribution that make a disease compatible with your framework? Specifiy.

Throughout: "principal components" not "principle components"

Reviewer #3: Jiang and colleagues studied the time varying effect of polygenic risk on 24 common ICD-10 codes with a prevalence of at least 1.5%, within the British ancestry subset of UK Biobank. The set of common disease codes analyzed each had ~10,000 cases in the UK Biobank and the authors matched each case with four controls. They used a proportional hazards model with an interval-based censoring to estimate genetic risk in different age groups. They found age varying risk profiles for 9 diseases including hypertension and atherosclerotic heart disease and for some diseases there were distinct profiles of genetic risk. For most diseases, polygenic risk had the most impact early on with monotonic decrease over time. Using simulation and permutation, the authors proposed several possible models that might explain their observations.

Comments

There really are not ‘24’ diseases but 24 distinct ICD-10 codes. Several related ICD-10 codes were used for various manifestations of the same disease eg atherosclerotic coronary heart disease (MI, unstable angina etc). These ICD codes could have been collapsed into phecodes to increase power to detect the genetic risk variability over time and improve readability.

Only in 9 of the 24 ‘diseases’ was there variability of genetic risk with age. In the remaining 14, is the lack of variability due to lack of power to detect variability? If not, how do these diseases fit into the framework of genetic risk, liability and frailty outlined by the authors?

It would be helpful to know whether estimates of heritability change with age in the various age groups. Along these lines, the impact of family history on the risk of developing disease at different age groups would be of interest.

In GWAS, we estimate the effect size within the age group of cases. GWAS participants are typically middle-aged individuals (mean age 62 years). Application of these risk estimates to much younger individuals introduces an element of imprecision. (The same can be said of much older individuals eg 75-80 y). The authors should comment on this.

Several of the diseases are related e.g., Hypertension and Hypercholesterolemia are risk factors for CHD which in turn is the precursor for Left Ventricular Failure. It would be of interest to assess how these associations impact the modeling of genetic risk effect with age.

The authors attempt to account for unmeasured risk factors on risk estimates. How about known risk factors? For example, for CHD it would be of interest to assess polygenic risk after adjustment for related covariates (for example hypertension and diabetes).

Do the analyses account for sampling of cases in the case control analysis? Population, age and sex-specific incidences rate could be used to account for how cases and controls were sampled using appropriate sampling weights.

The authors partition age between 45-75 by 5-year increments. The partitioning could have been informed by where the baseline hazard rate changes for different diseases. While this may be difficult to do for all diseases, it could be done for one disease, as an example.

Polygenic risk is more strongly associated with disease onset at younger ages; however, the incidence of disease is typically much lower and absolute risk may remain low even though polygenic risk is high. This has implications for clinical use. The authors should comment.

**Have all data underlying the figures and results presented in the manuscript been provided?**

Reviewer #1: Yes

Reviewer #2: Yes

Reviewer #3: Yes

PLOS authors have the option to publish the peer review history of their article (what does this mean?). If published, this will include your full peer review and any attached files.

Reviewer #1: No

Reviewer #2: **Yes: **Peter Kraft

Reviewer #3: No

---

## [Decision Letter · Decision Letter 1]

16 Jul 2021

Dear Dr McVean,

We are pleased to inform you that your manuscript entitled "The impact of age on genetic risk for common diseases" has been editorially accepted for publication in PLOS Genetics. Congratulations!

We do have two minor clarifying requests, however:

1. Suppl Fig 6A: Please explain in the legend what the blue lines are in the last two graphs and perhaps clarify why the other graphs don't have them. 

2. Suppl Fig 7: Please explain in the legend why there are only 20 graphs rather than 24 for all diseases, and why the ninth panel, between "Secondary malignant neoplasm of liver" and "Arthrosis, unspecified," is blank.

Yours sincerely,

Teri Manolio

Guest Editor

PLOS Genetics

David Balding

Section Editor: Methods

PLOS Genetics

Comments from the reviewers (if applicable):

Reviewer #3: The reviewers have adequately responded to the critique.

**Have all data underlying the figures and results presented in the manuscript been provided?**

Reviewer #3: Yes

PLOS authors have the option to publish the peer review history of their article (what does this mean?). If published, this will include your full peer review and any attached files.

Reviewer #3: **Yes: **Iftikhar J Kullo

**Data Deposition**

http://datadryad.org/submit?journalID=pgenetics&manu=PGENETICS-D-21-00007R1

**Press Queries**

---

## [Editor Report · Acceptance letter]

6 Aug 2021

PGENETICS-D-21-00007R1 

The impact of age on genetic risk for common diseases 

Dear Dr McVean, 

We are pleased to inform you that your manuscript entitled "The impact of age on genetic risk for common diseases" has been formally accepted for publication in PLOS Genetics! Your manuscript is now with our production department and you will be notified of the publication date in due course.

With kind regards,

Andrea Szabo

PLOS Genetics

On behalf of:
